# Isolation of pseudocapacitive surface processes at monolayer MXene flakes reveals delocalized charging mechanism

Marc Brunet Cabré [1], Dahnan Spurling[1], Pietro Martinuz[1,2], Mariangela Longhi [2], Christian Schröder [1], Hugo Nolan [1], Valeria Nicolosi [1], Paula E. Colavita [1] & Kim McKelvey [1,3] ✉

Pseudocapacitive charge storage in $Ti_3C_2T_x$ MXenes in acid electrolytes is typically described as involving proton intercalation/deintercalation accompanied by redox switching of the Ti centres and protonation/deprotonation of oxygen functional groups. Here we conduct nanoscale electrochemical measurements in a unique experimental configuration, restricting the electrochemical contact area to a small subregion ($0.3\ \mu m^2$) of a monolayer $Ti_3C_2T_x$ flake. In this unique configuration, proton intercalation into interlayer spaces is not possible, and surface processes are isolated from the bulk processes, characteristic of macroscale electrodes. Analysis of the pseudocapacitive response of differently sized MXene flakes indicates that entire MXene flakes are charged through electrochemical contact of only a small basal plane subregion, corresponding to as little as 3% of the flake surface area. Our observation of pseudocapacitive charging outside the electrochemical contact area is suggestive of a fast transport of protons mechanism across the MXene surface.

The transition to a low-carbon economy based on renewable energy requires the development of energy storage technologies. Supercapacitors, characterized by both high-power density and high-energy density, bridge the gap between rechargeable batteries and more traditional parallel-plate capacitors[1]. The development of new supercapacitor technology depends on the development of new materials, and this is supported by the precise understanding of the physical nature of the electrochemical charge storage mechanism[2–4].

MXenes are two-dimensional materials from the family of transition metal carbides, nitrides, and carbon-nitrides with the structure $M_{n+1}X_nT_x$ ($n = 1,2,3$)[5]. Among other applications[6], MXenes exhibit excellent performance as supercapacitors due to their high specific surface area, metallic-like conductivity, and pseudocapacitive response[7]. Titanium carbide MXenes ($Ti_3C_2T_x$) can be obtained by facile exfoliation, display high stability and allow several electrode

architectures, with specific gravimetric capacitances about $250\ F/g$[8]. The origin of charge storage in acidic media is fast ion intercalation into interlayer spaces coupled with the change in the oxidation state of the Ti and protonation of the oxygen functional groups ($T_x \rightarrow -O$ to $-OH$)[9–13]. Macroscale MXene electrodes, however, are complex 3D networks of individual MXene flakes, which affect ion transport from the electrolyte throughout the material network. As a result, on macroscale electrodes we can distinguish surface processes, which involve fast protonation of surface sites exposed to electrolyte and occur at shorter timescales, and bulk processes, which involve ion conduction and intercalation processes through the 3D network and occur at longer timescales[14].

In this study, we quantify the intrinsic electrochemical pseudocapacitive response of monolayer $Ti_3C_2T_x$ MXene by isolating the capacitive response on $0.3\ \mu m^2$ regions of monolayer $Ti_3C_2T_x$ MXene

[1]School of Chemistry, Trinity College Dublin, Dublin 2, Ireland. [2]Università degli Studi di Milano, Dipartimento di Chimica, Via Golgi 19, 20133 Milano, Italy. [3]MacDiarmid Institute for Advanced Materials and Nanotechnology, School of Chemical and Physical Sciences, Victoria University of Wellington, Wellington 6012, New Zealand. ✉e-mail: kim.mckelvey@vuw.ac.nz

flakes immobilized on a carbon supporting electrode using scanning electrochemical cell microscopy (SECCM)[15], as shown in Fig. 1. In our nanoscale SECCM configuration the bulk effects, that might arise from the macroscale 3D electrode, are eliminated and so any contributions from ion-intercalation processes. Therefore, the SECCM configuration allows us to isolate surface dependent processes that contribute to MXene pseudocapacitive response. Using a SECCM approach we measure cyclic voltammograms on a regular grid of sample points spaced 1.80 μm apart on a region of monolayer $Ti_3C_2T_x$ flakes. Cyclic voltammograms are acquired on both $Ti_3C_2T_x$ flakes and the surrounding carbon substrate, allowing us to compare the response on different flakes, different parts of the same flake, and control sample points of the carbon substrate.

## Results

### $Ti_3C_2T_x$ flake characterization

A stock dispersion of $Ti_3C_2T_x$ flakes was obtained by liquid exfoliation of MAX phase ($Ti_3AlC_2$). Freestanding films were prepared via vacuum filtration using stock $Ti_3C_2T_x$ dispersions (28 mg/ml) on which EDX, Raman, and XRD characterization were performed. Supplementary Figs. 1–3 show the XRD, EDX, and Raman spectra, which are consistent with those of $Ti_3C_2T_x$.

$Ti_3C_2T_x$ flakes were drop cast on a carbon surface, and isolated flakes were selected for electrochemical characterization using an SECCM approach. The morphology of individual $Ti_3C_2T_x$ flakes was determined by a combination of atomic force microscopy (AFM) and scanning electron microscopy (SEM), which indicates that the flakes are monolayer (see Supplementary Note 1).

### Localized electrochemical measurements on $Ti_3C_2T_x$ flakes

On our isolated $Ti_3C_2T_x$ flakes the backscattered SEM images show the electrolyte residues remaining after SECCM measurements with a total of 80 points identified (see Supplementary Fig. 7A). From the 80 sample points 64 points presented a well-defined circular geometry which allowed us to determine the electrochemical surface area (i.e., the geometric contact area defined by the SECCM droplet on the sample surface), which was found to be $0.31 \pm 0.03$ μm² (see Supplementary Fig. 8). As shown in Fig. 1c, a total of 24 points were found to partially or completely contact the MXene flake; of these, 5 were unambiguously located on the basal plane of monolayer $Ti_3C_2T_x$. 40 points were identified as contacting the carbon substrate exclusively (see Supplementary Note 2 for further details).

Representative voltammograms on the carbon surface and on the basal plane of monolayer $Ti_3C_2T_x$ flakes are shown in Fig. 2a. All ana-

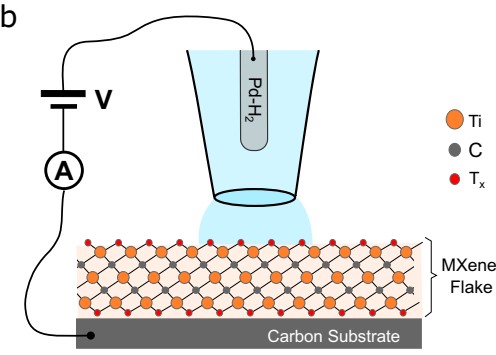

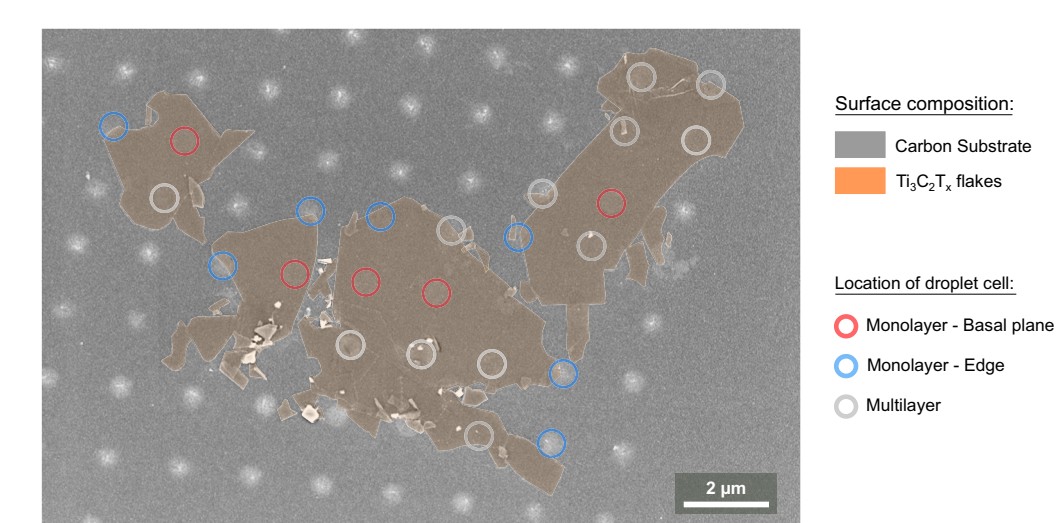

**Fig. 1 | Experimental configuration used to isolate monolayer MXene pseudocapacitive responses. a** Schematic of the SECCM configuration for measuring monolayer $Ti_3C_2T_x$ flakes immobilized on a carbon supporting electrode surface. SECCM-based cyclic voltammogram measurements were conducted in a hopping mode, with the probe movement pattern shown in coloured arrows (blue approach, red retract, black move to next measurement position). **b** Schematic of end of SECCM probe, highlighting the nanoscale electrochemical droplet cell at the end of the SECCM probe and the two-electrode electrochemical cell configuration. **c** Electron micrograph of sample surface containing monolayer MXene immobilized on a carbon surface after SECCM measurements, with each SECCM sample location highlighted according to the surface composition.

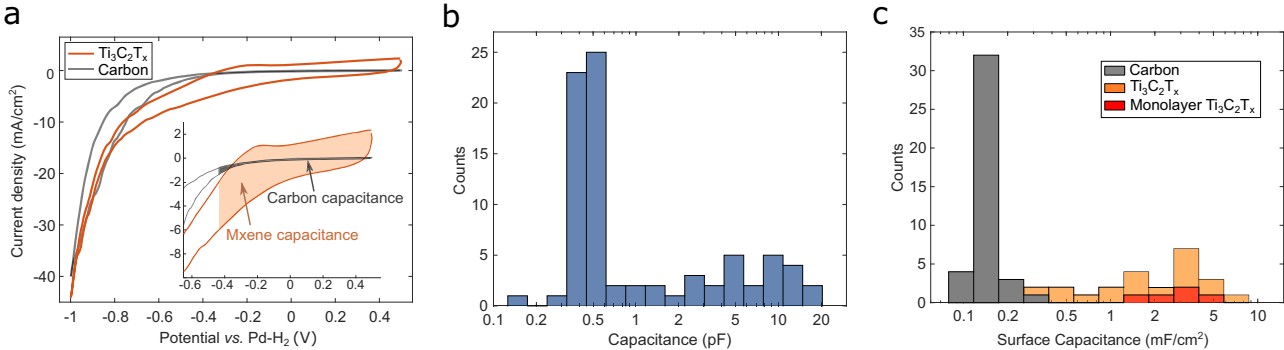

**Fig. 2 | Capacitive response on monolayer MXene flakes and surrounding carbon substrate. a** Representative cyclic voltammograms over a carbon surface (black) and a single monolayer MXene flake (orange) at scan rate of 0.5 V/s in 20 mM $HClO_4$. **b** Histogram of the capacitance at each individual SECCM grid point observed on the SEM image ($N = 80$). **c** Stacked histogram of the surface capacitance over carbon surface (black, $N = 40$) and MXene flake (orange, $N = 24$), of which the basal plane of single layer $Ti_3C_2T_x$ points are highlighted (red, $N = 5$).

lyses were carried out based on the second cycle of the CV response and all cyclic voltammograms obtained on carbon and $Ti_3C_2T_x$ can be found in Supplementary Note 2. Between +0.5 and −0.5 V vs Pd-$H_2$ (Pd-$H_2$ is +50 mV vs SHE) the voltammograms obtained over $Ti_3C_2T_x$ flakes, displayed in Fig. 2 and Supplementary Fig. 10, show the characteristic $i$–$V$ curves of pseudocapacitive charging in acidic media[10,16,17]. Below −0.6 V vs Pd-$H_2$ an exponential increase in the current magnitude is observed vs applied potential on both $Ti_3C_2T_x$ flakes and the carbon substrate, which is consistent with the onset of the hydrogen evolution reaction (HER)[6,18].

The potential window, −1 to 0.5 V vs Pd-$H_2$, was chosen to induce pseudo-capacitive and HER responses without inducing irreversible anodic oxidation, which occurs above +0.7 V vs Pd-$H_2$ (+0.75 V vs. SHE)[17]. We cycled into the HER response region to condition the MXene surface by saturating terminal oxide groups with adsorbed protons[12]. The first cycle over each point of the SECCM is considered as a conditioning step[10], and the capacitance response is determined from the second cycle.

Mechanical instability issues are common of macroscale MXene electrodes when placed under electrolyte[19]. The SECCM configuration, which only wets a very minor portion of the sample surface, prevents MXene flakes from lifting off from the surface. The SECCM droplet cell ensures rapid gas transport to the liquid–air interface to prevent bubble formation during hydrogen evolution[20,21]. The AFM and SEM, show that the MXene layers are intact on the carbon working electrode support and show no evidence of exfoliation.

## Observation of capacitive responses on subregions of $Ti_3C_2T_x$ flakes

The capacitance was determined by integrating the charge between +0.5 and −0.5 V vs Pd-$H_2$, as illustrated in Fig. 2a. A histogram of the capacitance values obtained for all SECCM grid points ($N = 80$) is displayed in Fig. 2b and suggests the presence of two distinct populations. To account for different contact areas, the capacitance was normalized by the geometric area to derive the specific surface capacitance at each point. Furthermore, the location of each SECCM grid point, determined from the electrolyte residues in SEM images, allowed us to correlate the capacitance at each point to the morphology of the surface contacted, i.e., only carbon contact ($N = 40$) and partial/complete $Ti_3C_2T_x$ flake contact ($N = 24$). Figure 2c displays stacked histograms of the specific surface capacitance obtained at carbon contact points and at MXene flake contact points, with points contacting a monolayer basal-plane of $Ti_3C_2T_x$ exclusively highlighted in red (see Supplementary Note 2). The average surface capacitance obtained for carbon was $0.15 \pm 0.04$ mF/$cm^2$, consistent with graphitic carbons in acidic electrolytes (up to 0.35 mF/$cm^2$)[22,23]. The average

surface capacitance measured on monolayer $Ti_3C_2T_x$ MXene sample points, exclusively, is $2.8 \pm 1.0$ mF/$cm^2$, more than an order of magnitude larger than that of the carbon support. As shown in Fig. 2c, the remaining points in contact with $Ti_3C_2T_x$ flakes present a distribution of specific surface capacitance, with values larger than the mean carbon specific surface capacitance. A detailed assignment of these points, shown in Supplementary Fig. 7, suggests that the broad distribution in specific surface capacitance is due to a wide range of flake morphologies (e.g., edge, multilayer).

MXene capacitive values derived from different approaches are often compared using gravimetric capacitance metrics. For instance, in experimental work carried out using macroscopic electrodes the capacitance is normalized by the mass of electrode material deposited over the geometric area contacted by the electrolyte[8,10,16,17,24−28]. Computational work translates monolayer simulations of specific areal capacitance into gravimetric capacitances[8,10,12,29−31]. In our SECCM experiments, we measure directly the electrochemical contacted area (see Supplementary Fig. 8). Assuming the crystalline structure of $Ti_3C_2T_x$, we can calculate the specific surface area of a monolayer single sided flake, $SSA_{1L\text{-one side}}$, as 272 $m^2 g^{-1}$ (see calculation details in Supplementary Note 3). Then, the equivalent mass of MXene contacted can be determined from the experimentally measured contact area. The electrochemical contacted area in our measurements is 0.31 $\mu m^2$, and therefore the mass of monolayer $Ti_3C_2T_x$ contained in the 0.31 $\mu m^2$ area is $1.15 \pm 0.10$ fg. We can normalize the capacitance values on monolayer points by this equivalent mass, yielding gravimetric capacitances between 4000 and 12,000 F/g for a monolayer basal plane. These values are remarkably high, one to two orders of magnitude greater than any previous theoretical prediction or measurement (see Supplementary Table 5)[24−26,32,33]. The $Ti_3C_2T_x$ pseudocapacitive charging is estimated to provide about 0.4 e- per unit cell per volt of storage when both sides of a monolayer are protonated[12,29,34]. A gravimetric capacitance of 12,000 F/g would be equivalent to 14.8 e- per unit cell per volt, an unphysically large capacitance that suggests that the MXene monolayer area engaged in capacitive charging is much larger than the area of the submicron droplet contact (0.31 $\mu m^2$).

SEM imaging shows that the MXene sample consists of four separate monolayer flakes (Fig. 3A). A comparison of the basal plane pseudocapacitance values ($N = 5$) obtained on the four flakes reveals differences, as shown in Fig. 3b, with a trend of increasing capacitance with increasing flake size. When the basal-plane capacitance values are normalized by the mass of the entire flake (see Supplementary Table 3), the specific gravimetric capacitance values are found to be independent of flake size and range between 180 and 300 F/g (see Fig. 3c). These estimates of gravimetric capacitance are in

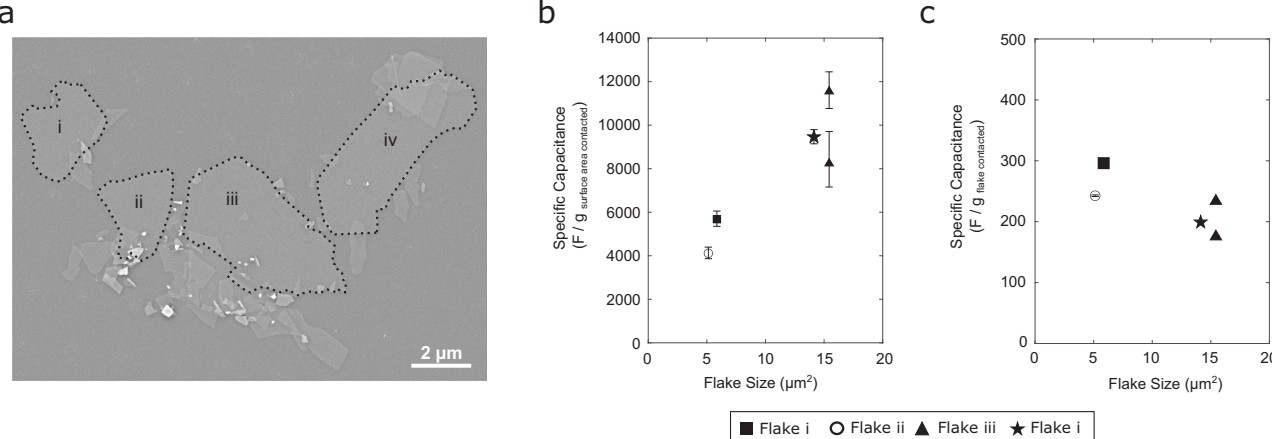

**Fig. 3 | Monolayer MXene capacitive response dependence on flake size.**
**a** Electron micrograph of sample surface with flakes larger than 5 µm² labelled.
**b** Specific gravimetric capacitance on monolayer $Ti_3C_2T_x$ compared to the total area $Ti_3C_2T_x$ flake, obtained by normalizing capacitance by the mass of MXene in electrochemical contact with electrolyte. **c** Specific gravimetric capacitance obtained by normalizing by the total mass of the monolayer flake. Error bars represent the standard deviation from determination of **b** MXene mass contacted and **c** MXene flake mass.

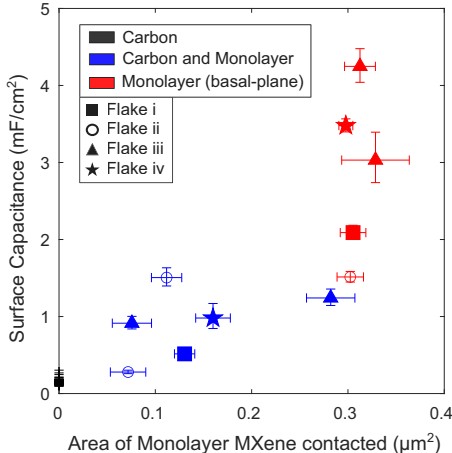

**Fig. 4 | Capacitance on edge and basal plane of monolayer MXene flakes.** Surface capacitance plotted against the area of $Ti_3C_2T_x$ monolayer contacted. Different symbols indicate which flake was contacted, while colour code indicate surface type. Data shows the mean and error bars represent the standard deviation.

excellent agreement with values predicted by DFT simulations (*ca.* 230 F/g)[12,29–31,34,35] and previous experimental determinations (220–250 F/g)[27,36]. Normalizing the basal-plane capacitance values by the two-sided area of the entire monolayer flake, we obtain specific surface capacitance values of $40 \pm 10$ µF/cm², which agrees with DFT prediction for $Ti_3C_2T_x$ of 45 µF/cm² [27]. This suggests that the capacitance response arises from the entire MXene flake and is not confined to the contact area between the MXene basal plane and our SECCM-based electrochemical cell.

**Implications for MXene pseudocapacitive mechanism**

In acid electrolytes the pseudocapacitance of MXenes is described as arising from proton intercalation/deintercalation accompanied by redox switching of the Ti centres and protonation/deprotonation of oxygen functional groups[4,11]. However, our samples consist of monolayer MXene on a carbon surface, and therefore ion intercalation would need to occur between the monolayer MXene and the underlying carbon surface. Further, we conduct our electrochemical measurements by establishing electrochemical contact with only a fraction of the basal plane of the MXene, which leaves no clear pathway for

intercalation of ions between the MXene and the carbon substrate. Nonetheless, we appear to be measuring the pseudocapacitance response from the entire MXene flake, despite our experimental configuration only allowing ion transport to approximately 3% of the total MXene flake surface.

Ion intercalation might be possible when contacting the boundary between the flake edge and the carbon substrate, and this could potentially provide an enhanced capacitive response[10,24,37,38]. On such sample points, the SECCM-based electrochemical cell is in contact with the carbon substrate-monolayer MXene gap, which could enable intercalation between the monolayer MXene and the supporting carbon surface. However, as shown in Fig. 4, edge points show a capacitance value per area of MXene contacted that is smaller than that at basal-plane monolayer points. This analysis suggests that ion intercalation at flake edges is not likely to be responsible for the specific pseudocapacitive values shown in Fig. 3.

## Discussion

In literature, capacitive *I–V* curves obtained on macroscopic 3D electrodes are often deconvoluted into current contributions from both surface and bulk processes[14,16,39], using a model described by Dunn et al.[40]. Both surface and bulk processes are important for describing the MXene pseudocapacitive behaviour, and it is useful to differentiate between the timescales of the fast protonation kinetics of $T_x$ groups (surface processes) and the contribution from the slower ion-intercalation (bulk processes)[14]. The Dunn et al. model however only considers two possible charging processes and assumes that the transport of charged species, i.e., bulk processes, can be described as a one-dimensional linear diffusion process[40]. This is a limitation for the description of 3D hierarchical structures that display intercalation. Deconvolution can in principle be improved by including other transport mechanisms[41,42]. However, increasing the number of parameters makes the modelling complex and can potentially lead to non-unique solutions[14].

The system studied here does not resemble any of the previous macroscale configurations used for MXene pseudocapacitance studies and enables measurement of discrete monolayer flakes without confounding effects that might arise from the 3D electrode architecture/organization. This enables the unambiguous assignment of the observed charging behaviour to surface processes, going well beyond approaches that involve mathematical deconvolution with a multi-parameter model. The results of our study show that by establishing electrochemical contact with only a small portion (approx. 3%) of basal

plane of a single monolayer MXene flake, the pseudocapacitive response observed is equivalent to that from the entire MXene flake. Assuming that pseudocapacitive behaviour in MXene monolayers is associated with protonation/deprotonation, our results suggest that protons are transported from/to the electrochemical cell over the entire MXene flake. Therefore, while our unique SECCM configuration isolates the surface processes and restricts ion-intercalation mechanisms, the proton transport effects are still found to dominate the capacitive response. Significantly, we conduct our measurements at 0.5 V/s, thus probing timescales where prior MXene descriptions stated that the response should be dominated by surface capacitive storage. Our results, however, identify that proton transport is likely to be present even at those very short timescales, where the MXene capacitive response is often thought to be independent of ion transport processes.

We speculate that the pseudocapacitive charging (i.e., $-O \rightarrow -OH$ surface protonation) outside the wetted area arises from the surface diffusion of protons in a water adlayer on the MXene flake surface[28,43,44]. Although we cannot exclude other possible proton transport mechanisms, such as proton transfer between functional groups ($-O$ and $-OH$ groups), proton tunnelling through the MXene layer or proton conduction through structural defects in the MXene[45]. Our measurements were conducted without atmospheric control (approx. humidity of $47 \pm 4$ %RH, see Supplementary Note 4), and it is likely that a thin water layer is present on the MXene surface that would facilitate proton transport. The AFM step-height profile of $Ti_3C_2T_x$ flakes suggests the presence of water adsorbed on its surface and/or water trapped between the carbon substrate and the $Ti_3C_2T_x$ flake (see Supplementary Note 1). The timescale of the cyclic voltammograms obtained in this work is on the order of 1 s; assuming surface diffusion of protons in a thin water layer, this would suggest that diffusion coefficients $>10^{-8}$ cm$^2$ s$^{-1}$ would be needed to access a 10 μm$^2$ flake surface during the electrochemical measurements. This is not an unreasonable diffusion rate, based on studies of proton dynamics at hydrophilic surfaces that reveal high proton mobility/diffusivity via water-assisted and anhydrous mechanisms[37,38,46–48].

The proton transport across the MXene surface at diffusion coefficients $>10^{-8}$ cm$^2$ s$^{-1}$ would act as a complementary mechanism supporting the retention of capacitive behaviour observed at ultrafast charging/discharging rates ($>1000$ V/s) for engineered three-dimensional networks[8,16]. Whereby, even limited percolation contacts might be sufficient to achieve very high specific gravimetric capacitances. Finally, these results suggest that MXene-based supercapacitors need to account for short time proton transport contributions, complementing the proton intercalation/deintercalation into MXene interlayer spaces.

## Methods

### Chemicals
Perchloric Acid (HClO$_4$, Fluka Analytical, 67–72%) was used as supplied by the manufacturer. All solutions were made with distilled Millipore water with a high resistivity of 18 MΩ cm. All procedures were carried out at room temperature.

### Preparation of carbon substrates
Carbon substrates were synthesized on SiO$_2$/Si wafers substrates via sputtering deposition followed by graphitization under inert atmosphere. SiO$_2$/Si wafers (300 nm thermal oxide) were first cleaned with piranha solution (3:1 H$_2$SO$_4$/H$_2$O$_2$ CAUTION: Piranha solution is a strong oxidant which may react explosively with organic solvents and must always be used in a fumehood), then rinsed with Millipore water and dried under nitrogen prior to sputter deposition. Deposition was carried out as previously reported[48]; briefly, amorphous carbon thin films were deposited in a dc-magnetron sputtering chamber (Torr

international, Inc.) using a graphite target at a base pressure $<2 \times 10^{-6}$ mbar for 40 min using Ar as deposition gas (50 sccm, $1-2 \times 10^{-2}$ mbar). Films were subsequently graphitized at 900 °C in a tube furnace (Carbolite Gero) under N$_2$ flow for 60 min, yielding $73 \pm 3$ nm thick carbon electrodes.

### Preparation of Ti$_3$C$_2$T$_x$ stock solution
20 ml of 9 M HCl (Sigma) was added in a PTFE vented vessel containing 1.6 g of LiF powder (Sigma). To allow dissolution of LiF powder, the solution was stirred at 400 rpm for 10 min while the vessel was placed in a 35 °C oil bath. Keeping the vessel in the oil bath while stirring the solution, a total of 1 g of MAX Ti$_3$AlC$_2$ phase (Carbon-Ukraine ltd.) was added to the solution in small fractions, allowing the temperature to stabilize between additions and minimizing overheating of the solution. To achieve a complete etching of the MAX phase, the solution was kept at 35 °C and stirred at 400 rpm for 24 h. After this time, the solution was diluted with deionized water and centrifuged for 5 min at $2800 \times g$ (5000 rpm). The supernatant was discarded, the sediment was redispersed in deionized water and centrifuged again for 5 min at $2800 \times g$ (5000 rpm). This process was repeated until the solution was at pH 6. The solution was then vortexed for 30 min to ensure delamination of multilayer Ti$_3$C$_2$T$_x$ flakes into monolayer Ti$_3$C$_2$T$_x$ flakes. After vortexing the solution was centrifugated for 30 min at $250 \times g$ (1500 rpm), and the supernatant which contained the monolayer flakes was collected. A final centrifugation step for 1 hour at $2800 \times g$ (5000 rpm) was used to concentrate the monolayer flakes in the sediment, which was redispersed to obtain a stock solution of Ti$_3$C$_2$T$_x$ flakes of 4 g/ml. The Ti$_3$C$_2$T$_x$ synthesis method described here was previously reported[49].

### Preparation of monolayer MXene flakes supported on carbon electrodes
The stock solution was further diluted with distilled water down to 10 μg/ml. The Ti$_3$C$_2$T$_x$ stock and aliquots were bubbled with argon to degas the solution and the flask was filled with argon to store solution in an inert atmosphere. 2 μl of diluted solution were drop-cast onto carbon substrates within 24 h of obtaining the stock solution. The sample was left to dry overnight in air, obtaining regions within the drop-cast area with single MXene flakes on the carbon substrate, which established the bottom-contact connection. Electrochemical measurements were carried out within 1 day.

### Instruments
Optical, AFM, and SECCM measurements of monolayer MXene flakes supported on carbon electrodes were acquired on a Park NX10 (Park Systems, South Korea). The AFM images were obtained in a non-contact mode (NCM) with a PPP-NCHR cantilever type (force constant $= 42$ N/m, resonance frequency $= 330$ kHz, Nanosensors). AFM and SECCM measurements were done in a room with temperature control. The temperature and humidity inside the SECCM and AFM Faraday cage were recorded for 7 days (see Supplementary Note 4), with a mean temperature of $22.6 \pm 0.2$ °C and relative humidity between 40 and 60 %RH. SEM images were acquired with a ZEISS Ultra Plus field-emission SEM with the secondary electron detectors, SE2 and In-Lens, at acceleration voltage of 3 kV. Energy dispersive X-ray spectroscopy (EDX) was performed on Zeiss Ultra Plus field-emission SEM at an acceleration voltage of 10 keV with a 20 mm$^2$ Oxford Inca EDX detector. X-ray diffraction (XRD) was obtained using the powder diffractometer Bruker D8 Discovery, in θ/2θ configuration and range of 3–75° at 2° min$^{-1}$. Raman spectroscopy measurements were acquired using a WITec Alpha 300R with a 633 nm He-Ne laser source and 1800 lines/mm grating. The structural characterization measurements (EDX, Raman, and XRD) were performed on as-synthesized Ti$_3$C$_2$T$_x$ thin film produced by vacuum filtration.

## Probe preparation

SECCM probes were single-barrelled nanopipettes with approximately 400 nm aperture radius. The nanopipettes were fabricated from single-barrelled borosilicate capillaries (1.5 mm O.D and 0.86 mm I.D., BF150-86-7.5, Sutter Instrument, USA) using a P-2000 laser puller (Sutter Instrument, USA). Using a pipette filler (MicroFil MF34G-5, World Precision Instruments, USA) the nanopipette was filled with 20 mM $HClO_4$ electrolyte. A Pd-$H_2$ quasi reference counter electrode (QRCE) was inserted at the top end of the pipette; prior to this, a Palladium wire (0.25 mm diameter, 5 cm long, PD005130, Goodfellow, UK) was biased at −3 V vs. a Pt counter electrode in 20 mM $HClO_4$ solution for 15 min to yield the Pd-$H_2$ quasi reference electrode[50,51]. Pd-$H_2$ QRCE was calibrated against the standard calomel electrode (SCE) after the SECCM scan with a value of −191 mV, which corresponds to a potential of +50 mV vs Standard Hydrogen Electrode (SHE).

## Scanning protocol

Electrochemical SECCM measurements were performed over a sample region where monolayer $Ti_3C_2T_x$ flakes were immobilized, as identified using optical microscopy (see Supplementary Fig. 4). SECCM imaging was carried out on a regular grid of sample points spaced 1.8 μm apart. At each SECCM sample point two cyclic voltammograms were measured between +0.5 and −1 V vs. Pd-$H_2$ at a scan rate of 0.5 V/s. Cyclic voltammograms were acquired over both $Ti_3C_2T_x$ flakes and the surrounding carbon substrate, as we can see from the salt residues shown in Fig. 1c. A hopping mode was used in which the probe was approached vertically towards the sample surface at a speed of 0.2 μm/s and a potential of −0.5 V was held until contact between the nanopipette droplet and the surface was established. The contact was detected as the appearance of a double layer charging current, which exceed a defined absolute threshold current of 3.0 pA. After approach, the potential was changed to +0.5 V and after a holding time of 2.0 s, two voltammetry cycles were recorded; then the pipette was retracted and moved to the next sample point of the pre-defined grid. Note, SECCM scans leave droplet residues on the surface, and when using $HClO_4$, the droplet cell residues were smaller than using $H_2SO_4$. The resulting small morphological features enabled AFM scanning to resolve the monolayer and bilayer MXene flake steps in the sample region.

## Data availability

The authors declare that all data supporting the finding could be found in the manuscript and supporting information. Raw datasets obtained from electrochemical and morphological characterization are available anytime upon request to the corresponding author.

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

## Acknowledgements

We acknowledge Park Systems for loan of a Park NX10 instrument. M.B. acknowledges the School of Chemsitry, Trinity College Dublin for PhD funding. This project has received funding from the European Union's Horizon 2020 research and innovation programme under the Marie Skłodowska-Curie Grant Agreement No. 713567 (EDGE-Project HECAT4H2), received by H.N. The results of this publication reflect only the authors' view and the Commission is not responsible for any use that may be made of the information it contains. This publication has also emanated in part from research conducted with the financial support of Science Foundation Ireland under Grant No. 19/FFP/6761, received by P.E.C. Authors thank the support of European Research Council (ERC) CoG, 3D2DPrint (GA 681544), received by V.N. K.M. acknowledges the support of the MacDiarmid Institute for Advanced Materials and Nano-technology. P.M. acknowledges Erasmus Traineeship Project. SEM imaging was carried out at the Advanced Microscopy Laboratory (AML) at the AMBER Research Centre, Trinity College Dublin, Ireland.

## Author contributions

M.B.C. and P.M. performed SECCM experiments and optimized sample preparation. D.S. designed the exfoliation process and manufactured the Mxene flakes. M.B.C. and D.S. performed physical characterization by AFM and SEM. C.S. and H.N. designed and manufactured carbon substrates. K.M., P.E.C., V.N., and M.L. conceived and designed the project. M.B.C., K.M., and P.E.C. analyzed the data and wrote the manuscript.

## Competing interests

The authors declare no competing interests.
