## [Peer Review File · Nature Communications]

Isolation of pseudocapacitive surface processes at monolayer MXene flakes reveals delocalized charging mechanismREVIEWER COMMENTS

Reviewer #1 (Remarks to the Author):

Authors have reported on the pseudocapacitive properties of MXene monolayers, investigated in a very interesting way. In general, I find the paper important and bringing scientific novelty of high importance for energy storage field, however, before recommending the paper for acceptance, I would like the Authors to address following points:

1. The electrolyte selection is not clear. Considering the application in electrochemical capacitors (please avoid 'supercapacitor' term), one would expect more common solution like H₂SO₄ or Li₂SO₄/LiNO₃-based formulations. Please describe the motivation for selecting your electrolyte.
2. The cyclic voltammograms indicate that there is electrolyte decomposition during scans. Are the results the same when the vertex potential does not exceed the HEP values? The recent results might suggest that there is an increase in capacitance because nascent hydrogen could 'exfoliate' the material.
3. I do not understand the translation from fg scale to g-scale. This could be a source of serious mistake and has nothing to do with real values (12 000 F/g looks enormously high). In the real construction, the bulk material will not allow to reach so high values, as the outer space (or just the interface) will be much smaller. I think that F/cm² metric is enough.
4. I would expect the EIS spectrum at certain potential values in order to see whether the increase of capacitance has capacitive or redox-based origin. Authors could provide any other results that allow for clear determination of capacitive and faradaic origin of the charge.
5. I realise that the MXenes are unstable at high potentials when investigated in larger cells. Is it the case for the samples investigated by Authors?

Having these points correctly addressed, I can consider my support for manuscript acceptance.

Reviewer #2 (Remarks to the Author):

The paper by Cabre et al reports on the unique electrochemical measurements on MXene monolayers. The experiment done here is complex but the authors have done commendable work to get the results that show high specific capacitance from the CV measurements.

The reviewer feels though that the comparison done in Table S5 will only make sense if the masses of the compared devices are within the same range because the masses of these monolayer flakes are very small and hence translate to such a high specific capacitances.

It recommended that the authors perform charge- discharge measurements and also the stability.

Reviewer #3 (Remarks to the Author):

The manuscript from Cabré et al reports on the first SECCM study on MXene, enabling local CV measurements on basal and edges sites of single MXene flakes. The experiments are carefully performed and well described. The main result is that even though small areas of MXene flakes are in contact with the electrolyte solution, the recorded capacitance can only be explained by proton diffusion to the entire flake. While I find this result very interesting from a technical perspective, I doubt that this will revisit the current understanding of pseudocapacitance in MXenes.

By definition, pseudocapacitance is governed by surface redox process. It is already well-known that for MXene, especially at high cycling rate (500 mV/s in this work), surface processes dominate the capacitance (see for example Shao et al, Energy Storage Materials 2019, 18, 456-461, showing that 80% of capacitance is due to surface processes at 100 mV/s). Proton de/intercalation processes may only play a dominating role at cycling rates at least 50 times smaller than used in this work.

Furthermore, the strategy mentioned in the last sentence suggesting optimization of MXene surface accessibility has already been implemented by many groups since at least 5 years (see reference 15

and follow-up papers).

I therefore do not believe that the novelty of the results is high enough to justify publication in Nature Communications. That being said, this work is of high relevance for a more specialized audience since it opens the field of SECCM on MXenes. I would suggest to consider the following points before resubmitting it to another journal:

- Do the author have any spectroscopic measurement (for example Raman) providing chemical information of the investigated MXene? I assume that HF-etched MXene were used therefore the presence and relative amount of F-terminated group should be mentioned somewhere in the main text (not only SI) as it will affect the number of redox sites.
- The CVs were run at quite negative potential (-1V vs SHE), which shows a drastic increase in current density, suggesting that HER is taking place in this regime. How does this affect the MXene surface chemistry? Can this explain the different CVs profiles between 1st and 2nd cycles?
- The author mention that no atmospheric control was used but can they still provide an estimate of the relative humidity of the cell environment before droplet formation?
- The reference to gravimetric capacitance of 12000 F/g in the abstract is misleading. I am not convinced by the relevance of using gravimetric values for a material not fully immersed in an electrolyte. As explained later in the text, this number is unphysical and has to be revised considering the full flake volume. I would remove the gravimetric values and only refer to surface capacitance to avoid misunderstanding.
- There are a few typos in the manuscript, including the title.

Reviewer #1:

Authors have reported on the pseudocapacitive properties of MXene monolayers, investigated in a very interesting way. In general, I find the paper important and bringing scientific novelty of high importance for energy storage field, however, before recommending the paper for acceptance, I would like the Authors to address following points:

Q1. The electrolyte selection is not clear. Considering the application in electrochemical capacitors (please avoid 'supercapacitor' term), one would expect more common solution like H₂SO₄ or Li₂SO₄/LiNO₃-based formulations. Please describe the motivation for selecting your electrolyte.

Aqueous 20 mM HClO₄ electrolyte was chosen for our measurements because it facilitates the characterization of the electrolyte residue to clearly define the probed geometric area, while also enabling AFM characterization of the probed area after SECCM measurements. We found that when using HClO₄ as the electrolyte the droplet cell residues were smaller than using H₂SO₄ electrolyte. Smaller morphological features from droplet residues enabled AFM scanning to resolve the monolayer and bilayer MXene flake steps of the probed sample region.

To address this in the revised experimental section we have clarified that 20 mM HClO₄ solution was used as electrolyte because it facilitates the morphological characterization by AFM. See modifications in page 11, line 28 and page 12, lines 13-15

Importantly, our electrolyte choice does not constitute a limitation of our measurements. Acidic electrolyte solutions favour the redox response of the MXene terminal groups (T_x) which is at the origin of the MXene pseudocapacitive response. The same electrochemical behaviour is observed regardless of whether H₂SO₄ or HClO₄ are used. Several determinations of the specific capacitance of Ti₃C₂T_x have been obtained using H₂SO₄ [see ref. 25, 26, 32 and 36 of the revised manuscript.] and similar trends are observed when using HClO₄ as shown in [doi.org/10.1002/elan.202100269; doi.org/10.1039/c3cc44428g; doi.org/10.1021/acsami.0c02446; doi.org/10.1002/sml.202002888].

Q2. The cyclic voltammograms indicate that there is electrolyte decomposition during scans. Are the results the same when the vertex potential does not exceed the HEP values? The recent results might suggest that there is an increase in capacitance because nascent hydrogen could 'exfoliate' the material.

The electrolyte is HClO₄ for all experiments which is not amenable to decomposition. For the potential window used, +0.45 V to -1.05 V vs SHE, we observe both pseudocapacitive charging and hydrogen evolution. Anodic potentials were not high enough to achieve irreversible oxidation of the MXene flake which occurs above +0.75 V vs SHE. (DOI:10.1002/anie.201911604).

Conditioning of MXene electrodes via cycling is conventionally carried out prior to capacitance determinations. [https://doi.org/10.1021/acsenergylett.0c01290] The first scan includes a potential sweep into the HER region which, according to computational studies, saturates the MXene terminal groups with adsorbed protons. [https://doi.org/10.1021/acs.jpcl.8b00200]

Characterisation of regions using both AFM and SEM after SECCM probing shows that MXene layers remain intact at the carbon working electrode support and show no evidence of exfoliation. The area wetted by the SECCM droplet is too small to result in any MXene flake detachment or "lift-off" from the surface, as would instead be expected if the whole monolayer were immersed in electrolyte.

To address this a brief discussion of the different electrochemical processes taking place, and their relevant potential regions, is included in the revised version of the manuscript. Also, a brief discussion on how MXene stability is ensured when using SECCM small droplet cells, is also included in the first part of the results section. Modifications in page 4, lines 22-33.

Q3. I do not understand the translation from fg scale to g-scale. This could be a source of serious mistake and has nothing to do with real values (12 000 F/g looks enormously high). In the real construction, the bulk material will not allow to reach so high values, as the outer space (or just the interface) will be much smaller. I think that F/cm² metric is enough.

We agree with the reviewer that a 12000 F/g capacitance value is un-physical and the reviewer's comment indeed highlights the discrepancy between estimates of gravimetric capacitance obtained from macroscopic electrodes and from single monolayer determinations in this work.

The procedure used for estimating specific gravimetric capacitance from SECCM results follows standard protocols applied to determinations obtained using both macroscopic electrodes and computational models. On macroscopic electrodes, the specific gravimetric capacitance is calculated by measuring the total capacitance followed by normalization by the mass of electrode material deposited over the geometric area contacted by the electrolyte [Ref 10, 17, 24, 25, 26, 28, 32, 33 and 37 of the revised manuscript]. The same procedure is followed in our SECCM study and therefore there is no operational source of error in the calculation. The geometric contact area is determined from the electrolyte residue area observed via SEM, and the mass of MXene monolayer underneath this contact area is calculated from the lattice structure of a monolayer. As these values are calculated for monolayer nanostructures assigned from the SEM images, there is no ambiguity in terms of the mass contacted during the measurements. Notably, several computational works predict specific gravimetric capacitance values, even when these simulations only include monolayer clusters, using an identical approach to ours [Ref. 8, 10, 12, 29, 31 and 34 of the revised manuscript].

Capacitance depends on MXene area contacted, thus a small amount of MXene contacted should provide an equivalent small capacitive current. The contribution to the total capacitance from the carbon substrate is much smaller and negligible relative to that of the MXenes, as confirmed by control experiments; therefore, this also is unlikely to be a source of error in our determinations. The discrepancy originating from the 12 000 F/g estimate is, in fact, what leads us to conclude that in our measurements the area we are charging is much larger than the area of the submicron droplet contact (0.3 μm^2).

We believe that the use of gravimetric capacitance values is useful for bridging our measurements via nanoscale electrochemistry to results obtained with macroscopic electrodes or computational simulations. The use of areal specific capacitance is also reported in our work thus allowing for both types of comparisons as needed and appropriate. Therefore, to address the reviewer's comment in the revised version of the manuscript, we have clarified that the method to estimate gravimetric capacitance is identical to that used for state-of-the-art experimental and computational determinations. See modifications page 3, lines 20-31. We have also expanded on the discussion in the text focused on the significance, relevance and conclusions that can be drawn from the gravimetric capacitance estimates. See modifications page 8, lines 6-22 and page 9, lines 2-8 and 24-34.

Q4. I would expect the EIS spectrum at certain potential values in order to see whether the increase of capacitance has capacitive or redox-based origin. Authors could provide any other results that allow for clear determination of capacitive and faradaic origin of the charge.

We agree with the reviewer that EIS experiments are useful and that they are typically carried out on macroscopic electrodes. However, EIS experiments at the nanoscale present very significant challenges and in fact there are only few reports of such experiments in the literature. Our SECCM- measurements yield currents of 10's pA, and such currents constitute a major challenge in signal processing, noise level reduction and temporal resolution due to bandwidth limitations of the current amplifier for pA scale currents [<https://doi.org/10.1002/celc.202001083>]. To the best of our knowledge examples of EIS using SECCM are limited to two recent publications [<https://doi.org/10.1021/acs.jpcc.2c03807>], [<https://doi.org/10.1021/acs.analchem.1c02972>] focusing on very specific sample/redox reaction configurations. Nonetheless, it is possible to speculate on the origin of the charging mechanism observed at the nanoscale on the basis of work on MXene pseudocapacitance.

In acid electrolytes, surface (T_x group) protonation and Ti-center oxidation, accompanied by proton intercalation processes have been found to underpin pseudocapacitance. [<https://doi.org/10.1021/acsenergylett.0c01290>] [<https://doi.org/10.1021/acs.jpcclett.8b00200>]. Charging via surface protonation was observed even when cycling at extremely high scan-rates (>1000 V/s) [<https://doi.org/10.1038/nenergy.2017.105>] which indicates that very short time scales are needed for surface protonation processes to occur. Our voltammograms, obtained at 0.5 V/s, show very similar behaviour as prior experimental studies at 0.5 V/s [<https://doi.org/10.1021/acsenergylett.0c01290>, shown in SI] [<https://doi.org/10.1038/nenergy.2017.105>]. Our capacitive values observed for monolayer MXene flakes, 230F/g, agrees with theoretical predictions and prior experimental studies. [<https://doi.org/10.1021/acsenergylett.0c01290>] [<https://doi.org/10.1038/nenergy.2017.105>] Therefore, we are observing the same surface changing mechanism on our monolayer MXene flakes.

What is unique to our findings, is that we are observing the entire MXene flake can be engaged in the charging process ($15 \mu\text{m}^2$) despite the establishment of a very limited contact area with the electrolyte ($0.31 \mu\text{m}^2$). This behaviour suggests a surface proton conduction mechanism is dominating the pseudocapacitive response. Previous literature suggested mass transport or ion-intercalation effects do not play a role on the faradaic origin of the charge at short timescales. However, isolating the monolayer MXene flake response allows us to observe a proton conduction mechanism occurring and dominating capacitive response at short timescale (0.5V/s). Please, also see response to Reviewer #3.

To address this comment in the results section, where it states “Localized electrochemical measurements on $\text{Ti}_3\text{C}_2\text{T}_x$ flakes” we added reference [<https://doi.org/10.1021/acsenergylett.0c01290>]. *See page 3, line 19*

Q5. I realise that the MXenes are unstable at high potentials when investigated in larger cells. Is it the case for the samples investigated by Authors?

The cathodic limit of the potential window used in our work is in the HER region and therefore we expect H_2 gas evolution to be taking place at the cathodic end of our sweeps. In the case of macroscopic electrodes the evolution of gas can create mechanical instabilities that degrade or change the response over time. However, in the case of SECCM experiments, gas evolution takes place on a mono-/few-layer electrode and the gas can transport very rapidly to the droplet cell air interface, preventing bubble formation. If gas bubble formation were to occur within

the SECCM droplet cell a discontinuity on the current trace of the voltammograms would be observed, because of the associated change in contact area and reactant mass transport. This has previously been observed when doing HER experiments with other 2D materials using SECCM [<https://doi.org/10.1021/acs.analchem.1c02099>] but we did not observe such behaviour, which suggests that bubble formation does not occur under our experimental conditions.

To address this comment we have included a brief discussion regarding the expected effects of gas evolution in the revised version of the manuscript. **See modifications in page 4, lines 27-32**

Q6. Having these points correctly addressed, I can consider my support for manuscript acceptance.

We thank the reviewer for the attentive reading of our manuscript and for the constructive comments.

Reviewer #2:

The paper by Cabre et al reports on the unique electrochemical measurements on MXene monolayers. The experiment done here is complex but the authors have done commendable work to get the results that show high specific capacitance from the CV measurements.

Q1. The reviewer feels though that the comparison done in Table S5 will only make sense if the masses of the compared devices are within the same range because the masses of these monolayer flakes are very small and hence translate to such a high specific capacitances.

We thank the reviewer for the positive comments. The calculation of gravimetric capacitances is carried out using identical assumptions as those used in the literature when dealing with macroscopic electrodes or devices (please see also reply to Reviewer 1 – Q3 comment). The majority of MXene papers uses gravimetric capacitance as a figure of merit and therefore a calculation of the same allows for a direct comparison to existing reports. This approach is also adopted by computational groups when simulations for the determination of specific capacitance on monolayers are used to predict gravimetric specific capacitance values.

To address this comment we have clarified in the result section of the main text that our approach to calculate gravimetric capacitance is same as taken for macroscale electrodes or computational works. See modification in page 5, lines 20-31

Q2. It recommended that the authors perform charge- discharge measurements and also the stability.

We agree with the reviewer that performing charge-discharge cycling, e.g. up to 10000 cycles as in [<https://doi.org/10.1038/nenergy.2017.105>] is useful to evaluate stability. However, such measurements require hours of continuous measurements, which is not possible in a SECCM approach. SECCM probes need to be positioned in close proximity to the MXene surface (approx. distance of a radius of the probe ~ 500 nm). However, the piezoelectric positioners and associated mechanical components that control the SECCM probe drift over time due to thermal changes. Therefore, once the SECCM droplet cell is formed, probe-surface positions vary slowly over time. If this situation is extended over long periods of time the SECCM probe either crashes into the MXene sample or stretches the SECCM droplet until contact is lost. This limits the time the SECCM droplet cell can be stationary and does not allow charge-discharge stability measurements. In the measurement reported here the SECCM-probe was stationary for 12 seconds (2 cycles with 1.5 V potential window with 0.5 V/s scan rate). The short time we used ensures that even with drift of 2 nm/s the pipette will not crash, and droplet cell will not be distorted.

Reviewer #3:

The manuscript from Cabré et al reports on the first SECCM study on MXene, enabling local CV measurements on basal and edges sites of single MXene flakes. The experiments are carefully performed and well described. The main result is that even though small areas of MXene flakes are in contact with the electrolyte solution, the recorded capacitance can only be explained by proton diffusion to the entire flake. While I find this result very interesting from a technical perspective, I doubt that this will revisit the current understanding of pseudocapacitance in MXenes.

By definition, pseudocapacitance is governed by surface redox process. It is already well-known that for MXene, especially at high cycling rate (500 mV/s in this work), surface processes dominate the capacitance (see for example Shao et al, *Energy Storage Materials* 2019, 18, 456-461, showing that 80% of capacitance is due to surface processes at 100 mV/s). Proton de/intercalation processes may only play a dominating role at cycling rates at least 50 times smaller than used in this work. Furthermore, the strategy mentioned in the last sentence suggesting optimization of MXene surface accessibility has already been implemented by many groups since at least 5 years (see reference 15 and follow-up papers).

I therefore do not believe that the novelty of the results is high enough to justify publication in *Nature Communications*. That being said, this work is of high relevance for a more specialized audience since it opens the field of SECCM on MXenes.

We thank reviewer 3 for this comment; we fully agree that both surface and bulk processes are important to explain MXene pseudocapacitive behaviour and their contributions have indeed been studied using macroscopic electrodes with 3D structures. However, the system presented in our work does not resemble any of the previous macroscale configurations used for MXene pseudocapacitance studies and enables measurement of discrete monolayer flake without confounding effects that might arise from the 3D electrode architecture/organisation.

Previous work, including Shao *et al.*, typically deconvolutes capacitive I - V curves into current contributions from both surface and bulk processes using a model described by Dunn *et al.* [*J. Phys. Chem. C*, 111 (2007), pp. 14925-14931], which correlates the capacitive current (i_C) with the sweep rate (v) according to:

$$i_C (V) = k_1 v + k_2 v^{1/2}$$

k_1 and k_2 determined from best-fits of the current are then used to quantify the fractional contribution from surface ($b = 1$) and bulk diffusion-limited processes ($b = 0.5$). This approach only considers two possible charging processes and assumes that the transport of charged species, i.e. bulk processes, can be described as a one-dimensional linear diffusion process. This is a limitation for the description of 3D hierarchical structures that display intercalation. Deconvolution can in principle be improved by including other transport mechanism, e.g. transport through porous media and hemispherical diffusion. However, as well articulated in Shao *et al.*, increasing the number of parameters makes the modelling complex and can potentially lead to non-unique solutions while not necessarily offering greater insights on the physical processes at play during storage. It is therefore widely acknowledged that alternative approaches are needed to better understand the connection between physical process and current response.

In the case of MXenes, it is well known that T_x groups present fast redox kinetics, while experimental observations on macroscale MXene electrodes also show a slower capacitive behaviour which has been attributed to mass transport and ion-intercalation mechanism. The model above by Dunn et al. revealed that a mix of 0.5 and 1.0 b values satisfactorily describes the two characteristic timescales associated with charging. In our case we directly access the

response of a discrete monolayer in the absence of MXene stacking/assembly, thus enabling the unambiguous assignment of the observed charging behaviour to surface processes, going well beyond approaches that involve mathematical deconvolution with a multi-parameter model. Still, with this configuration we discover an anomalous proton transport behaviour of MXenes, so that only contacting a minor area with electrolyte ($0.3 \mu\text{m}^2$) we observe capacitances that can only be justified by engaging up to $15 \mu\text{m}^2$ of the MXene in the charge storage process. Therefore, we are observing a proton transport mechanism to dominate the capacitive response despite only enabling what previously has been described as surface processes, which Dunn et al. model was considered to be independent from mass transport effect. Importantly, we conduct our measurements at 0.5 V/s , thus probing timescales where prior descriptions stated that the response should be dominated by pure capacitive storage, but again, we are observing proton conduction mechanism dominating the capacitive response. We believe the discovery of this behaviour represents a breakthrough in our understanding of the capacitive response of MXene electrodes by showing that proton conduction is likely to dominate their response at even very short timescales.

To address the reviewer's comment we have expanded on the discussion of the novelty aspects of our work and included the summarized description of the rationale stated above in the main text. The title has also been modified to provide a better description of the relevance and scope of our findings. *See modifications across the main text including abstract (page 1, lines 21-24), introduction (page 2, lines 15-20 and 24-27) and discussion (page 8, lines 6-22 and page 9, lines 2-8 and 24-33)*

Q1: Do the author have any spectroscopic measurement (for example Raman) providing chemical information of the investigated MXene? I assume that HF-etched MXene were used therefore the presence and relative amount of F-terminated group should be mentioned somewhere in the main text (not only SI) as it will affect the number of redox sites.

The synthesis of MXenes follows prior published work. To address this comment in this revised version we have added a reference to previous synthesis and characterisation, details of the MAX phase used for exfoliation and included XRD diffractogram, EDX and Raman spectroscopy of $\text{Ti}_3\text{C}_2\text{T}_x$ flakes in the Supporting Information. *See modification in the main text, page 3, lines 12-17 and page 4 lines 1-6. See the addition of the spectrums in Supplementary Figures 1, 2 and 3. Description of the corresponding methods has been included in page 11 lines 13-19.*

Q2: The CVs were run at quite negative potential (-1V vs SHE), which shows a drastic increase in current density, suggesting that HER is taking place in this regime. How does this affect the MXene surface chemistry? Can this explain the different CVs profiles between 1st and 2nd cycles?

Conditioning of MXene electrodes via cycling is conventionally carried out prior to capacitance determinations. [<https://doi.org/10.1021/acsenergylett.0c01290>] The potential window used includes a potential sweep into the HER region which saturates the MXene terminal groups with adsorbed protons. We therefore consider that by the second cycle the pseudocapacitive behaviour is conditioned.

Stepping into HER regime will cause the adsorbed protons on MXene surface to get reduce to form H_2 , which at first instance can seem to indicate that charge storage on the forward cycle (charging) will be lost before the backward cycling (discharging). However, T_x groups present a very fast redox protonation kinetics, and on $\text{Ti}_3\text{C}_2\text{T}_x$ the kinetics of HER (Heyrovsky or Tafel

steps) are expected to be much slower than MXene surface protonation processes (Volmer step), [DOI: 10.1039/d0ta11735h]. Therefore, despite adsorbed proton turn to H₂ during HER regime, the re-protonation of the surface occurs immediately. In consequence, we expect equilibrium of protonated redox sites (T_x) to be reached at any potential during the second cycle. Therefore, stepping or not onto HER regime during the second cycle (i.e. after counter ions are displayed in first cycle) should not have an effect over capacitance values derived. It is also worth noticing charging and discharging currents have a similar absolute value, indicating symmetric charging and discharging processes. To address this comment in the revised manuscript, we have added a paragraph in the first section of results to clarify stepping on HER regime during the second cycle would not affect capacitive charge stored. See modification in main text page 4, lines 22-32.

Q3: The author mention that no atmospheric control was used but can they still provide an estimate of the relative humidity of the cell environment before droplet formation?

We agree with the reviewer that providing information about relative humidity during the measurement would be useful to compare with other experiment. The relative humidity and temperature measurements have been included in the Supplementary information; see Supplementary Note 4, Supplementary Fig. 12. In Method section of the main text, page 11 lines 9-12, we have also clarified the temperature and humidity levels.

Q4: The reference to gravimetric capacitance of 12000 F/g in the abstract is misleading. I am not convinced by the relevance of using gravimetric values for a material not fully immersed in an electrolyte. As explained later in the text, this number is unphysical and has to be revised considering the full flake volume. I would remove the gravimetric values and only refer to surface capacitance to avoid misunderstanding.

The use of the gravimetric capacitance is useful for bridging our measurements via nanoscale electrochemistry to results obtained with macroscopic electrodes, where the specific gravimetric capacitances are used as the main metric to characterise the electrode materials (see e.g. ref. 24-26 and 29-34 of the revised manuscript). We believe that the use of gravimetric capacitance is not misleading, as the calculation of our value follows the same assumptions adopted by experimental work carried out using macroscopic electrodes, where the total electrochemically active mass is assumed rather than known; and by computational work that translates monolayer simulations of specific areal capacitance into gravimetric capacitances, in the same approach that we take. Please see also our response to Reviewer 1/Q3 on the use of gravimetric capacitances.

However, we acknowledge that presenting the gravimetric capacitance estimate of 12000 F/g in the abstract of the manuscript might cause an inaccurate first impression on the reader. To address this we have therefore removed this value from the abstract (main text page 1, lines 17-18) and included further clarification in the main text (page 5, lines 20-31) as indicated in Reviewer 1/Q3.

Q5: There are a few typos in the manuscript, including the title.

We have proof-edited the manuscript as recommended.

REVIEWERS' COMMENTS

Reviewer #1 (Remarks to the Author):

Authors provided the manuscript "Isolation of pseudocapacitive surface processes at monolayer MXene flakes reveals delocalized charging mechanism" revised in respect to the comments received in the previous review.

I accept their explanations and I find their answers convincing in majority of points. Nevertheless, I do not agree with explanation provided for calculations of specific capacitance from so small samples. Of course, this is not a calculation error, however, from practical point of view there is no reason for doing it. I do not accept the argument that others are doing it too.

In my opinion, this part could be removed as it does not bring any information/value to the manuscript. As all my comments were addressed in the revised version, I do not have further comments on the file.

Reviewer #3 (Remarks to the Author):

The authors have responded to previous comments with care and have improved the clarity of the manuscript, especially by improving discussion based on previous model of surface pseudocapacitive processes and putting the gravimetric estimation more in context. I can now recommend to accept this revised manuscript.

Point-by-point response to reviewers

Reviewer #1:

Authors provided the manuscript "Isolation of pseudocapacitive surface processes at monolayer MXene flakes reveals delocalized charging mechanism" revised in respect to the comments received in the previous review. I accept their explanations and I find their answers convincing in majority of points.

Q1: Nevertheless, I do not agree with explanation provided for calculations of specific capacitance from so small samples. Of course, this is not a calculation error, however, from practical point of view there is no reason for doing it. I do not accept the argument that others are doing it too. In my opinion, this part could be removed as it does not bring any information/value to the manuscript. As all my comments were addressed in the revised version, I do not have further comments on the file.

Capacitance is an extensive property of electrochemical systems. To compare between different sized systems capacitance values need to be normalised, either as specific capacitance (F/g) or surface capacitance (F/cm²). Capacitance refers to the charge stored at the electrode-solution interface, and so surface capacitance is in principle the most natural capacitance unit to report. However, macroscale supercapacitor systems that have complex electrode geometries typically only report specific capacitance values. Any metric of capacitance must be comparable across different experimental and computational measurements.

In the manuscript we had reported both specific capacitance and surface capacitance values. The specific capacitance values reported have been calculated based on experimentally measured surface area, from which the mass of MXene is derived assuming the crystalline structure of monolayer Ti₃C₂T_x. **We have clarified this point the revised version of the manuscript. See page 4 line 118-122 and page 5 lines 123-124.**

After extensive discussion, we believe reporting both values give the reader a better understanding of our results and enables the reader to relate to this work. In addition, specific capacitance is central to our discussion and conclusion that we are charging a much larger area than the submicron droplet contact area (0.3 μm²). For these reasons, we keep both specific capacitance and surface capacitance values in the text.

We appreciate the positive comments of Reviewer #1 on the answers we provided in the revised version of the manuscript.

Reviewer #3:

The authors have responded to previous comments with care and have improved the clarity of the manuscript, especially by improving discussion based on previous model of surface pseudocapacitive processes and putting the gravimetric estimation more in context. I can now recommend to accept this revised manuscript.

We appreciate the positive comments of Reviewer #3 and their recommendation for accepting the manuscript.